# Early Parenteral Administration of Ceftiofur has Gender-Specific Short- and Long-Term Effects on the Fecal Microbiota and Growth in Pigs from the Suckling to Growing Phase

**DOI:** 10.3390/ani10010017

**Published:** 2019-12-20

**Authors:** Ursula Ruczizka, Barbara Metzler-Zebeli, Christine Unterweger, Evelyne Mann, Lukas Schwarz, Christian Knecht, Isabel Hennig-Pauka

**Affiliations:** 1University Clinic for Swine, Department for Farm Animals and Veterinary Public Health, University of Veterinary Medicine Vienna, 1210 Vienna, Austria ; Ursula.Ruczizka@vetmeduni.ac.at (U.R.); Christine.Unterweger@vetmeduni.ac.at (C.U.); Lukas.Schwarz@vetmeduni.ac.at (L.S.); Christian.Knecht@vetmeduni.ac.at (C.K.); Isabel.Hennig-Pauka@tiho-hannover.de (I.H.-P.); 2Institute of Physiology, Pathophysiology and Biophysics, Unit of Nutritional Physiology, Department of Biomedical Sciences, University of Veterinary Medicine Vienna, 1210 Vienna, Austria; 3Institute of Food Safety, Food Technology and Veterinary Public Health, Unit of Food Microbiology, Department for Farm Animals and Veterinary Public Health, University of Veterinary Medicine Vienna, 11210 Vienna, Austria; Evelyne.Mann@vetmeduni.ac.at; 4Field Station for Epidemiology, University of Veterinary Medicine Hannover, 49456 Bakum, Germany

**Keywords:** ceftiofur, antimicrobial, microbiota composition, fecal microbiome, growth performance, pig

## Abstract

**Simple Summary:**

Antibiotics are commonly used in prevention and therapy of bacterial diseases in pig production. Although the main target of antibiotics are the pathogenic bacteria, they often disrupt the commensal gut microbiota as a whole, leading to intestinal disturbances. These detrimental effects have been well established for oral administration of antibiotics, whereas knowledge about potential disturbing effects of single parenteral antibiotic treatments on the gut microbiota development is limited. In this research, the impact of a single antibiotic injection on the first day of life on the maturation of the fecal microbiome and host growth performance was evaluated from the suckling to the growing phase. Results showed that a single antibiotic injection early in life influenced the bacterial community development in the short- and long-term and that this disturbance in the bacterial community was sex-specific. Present results further demonstrated that changes in the bacterial ecosystem of the gut may impair the growth performance of the growing pig. Thus, the results of the present study emphasize the importance of a proper and strict use of antibiotics in swine herds.

**Abstract:**

Using ceftiofur during the first days of life is a common preventative strategy against several bacterial diseases in pig production. This study aimed to evaluate short- and long-term effects of early use of ceftiofur on the fecal microbiota development in suckling and growing pigs. Sixty-four piglets from eight litters were assigned to the antibiotic (AB; *n* = 32) or control group (control; *n* = 32). Twelve hours postpartum (day 0) AB piglets received an intramuscular injection of ceftiofur (5.0 mg/kg body weight) or a placebo. DNA was extracted from fecal samples collected on days 0, 12, 28, and 97 for deep-sequencing of the *16S rRNA* gene. The AB administration disturbed the maturational changes in the fecal microbiome, whereby effects were sex-specific. Sex-related differences in AB metabolism in females and males may have caused these diverging AB-effects on the fecal microbiota. Especially the loss of bacterial diversity and of certain taxa in female AB pigs may have contributed to the decreased body weight of these females on day 97 of life. Taken together, this study showed that an AB injection with ceftiofur 12 h postpartum markedly affected the successional changes in the fecal microbiota composition in male and female pigs, with long-term consequences for host performance.

## 1. Introduction

For more than 50 years, antibiotics (AB) were commonly used in prevention and therapy of bacterial diseases in livestock animals, such as the pig [1,2]. Although aimed to target mainly pathogenic bacteria, oral administration of AB affects the commensal gut microbiota as a whole [3,4,5,6]. For this reason, the administration of in-feed AB (e.g., olaquindox, oxytetracycline, and kitasamycin) early in life led to compartmentalized variations in the microbial communities along the porcine gastrointestinal (GI) tract with an increase in potential pathogenic bacteria, such as *Escherichia* spp. and *Streptococcus suis* [7]. These findings were supported by others, showing dramatic alterations in the overall microbial community structure caused by in-feed AB like carbadox, the combination of chlortetracycline, sulfamethazine, and penicillin [8], a mixture of olaquindox, oxytetracycline, kitasamycin [9], and tylosin [10]. Due to the fundamental role the gut microbiota plays in host metabolism, immune functions, and physiology [11], previous studies also reported long-term consequences of AB for the host, such as an altered immune programming and growth [5,12].

So far, only few studies investigated the impact of injectable AB on the gut microbiota [13,14]. Accordingly, a single intramuscular injection of amoxicillin on the first day of life increased potential pathogenic bacteria including *Shigella* spp., *Escherichia (E.) coli* and *Salmonella enterica* and suppressed *Lactobacillus sobrius* in colonic digesta [13]. The third generation cephalosporine ceftiofur is frequently used during the first days of live as a common preventive strategy against bacterial diseases e.g., polyarthritis, diarrhea, and meningitis [15,16]. Ceftiofur was developed primary for the therapeutic use in veterinary medicine and has a high in vitro efficacy against several pathogens e.g., *Pasteurella multocida* spp., *Actinobacillus* spp., *Streptococcus* spp., *Haemophilus* spp., and *Salmonella Choleraesuis* in pigs [17]. However, the impact of an early-life injection of ceftiofur on the bacterial colonization in newborn piglets has not been elucidated so far. Evidence for a modulatory effect of ceftiofur exists from growing pigs where two different parenteral preparations (ceftiofur crystalline free acid and ceftiofur hydrochloride) caused drastic AB-specific changes in the composition and diversity of the fecal microbiota [18]. Notably, the fecal microbiota of pigs that received the ceftiofur hydrochloride returned back to the initial microbiota 14 days after the treatment, whereas the bacterial community of pigs receiving the ceftiofur crystalline free acid were still divergent from the initial bacterial community [18]. In another study, a ceftiofur treatment decreased the *E. coli* population significantly during the first days post-treatment, though the ceftiofur-treated pigs shed more extended-spectrum cephalosporins (ESCs)-resistant *E. coli* strains compared to the non-treated animals [14].

Against this background, we hypothesized that a ceftiofur injection on the first day of life would alter the initial colonization of the GI tract of newborn pigs, which may have long-term consequences for the establishment of the commensal microbiota and bacterial diversity as well as the host performance. Therefore, the present study aimed to investigate the effect of an early, single intramuscular injection of ceftiofur crystalline free acid on the development of the bacterial microbiome in feces of the suckling, weaned, and growing pig and pig’s growth performance. 

## 2. Materials and Methods 

### 2.1. Ethical Statement 

All procedures for animal handling, care and treatment of pigs have been approved by the institutional ethics committee of the Vetmeduni Vienna and the national authority according to paragraph 26 of Law for Animal Experiments, Tierversuchsgesetz 2012—TVG 2012 (GZ-68.205/0124-WF/V/3b/2014).

### 2.2. Animals and Housing

The litters (Large White × Piétrain) of eight third parity sows were used in the present study and housed in a temperature-controlled farrowing room (20 ± 2 °C). Immediately after birth, piglets were clinically examined for overall health. Piglets were nursed by the sow without cross fostering and were offered a commercial piglet starter diet (Provimi, Rotterdam, Netherlands) as creep feed from day 10 of life. Piglets were weaned on day 28 of life. Two litters were housed together in one pen in the rearing unit. At the age of twelve weeks, pigs were moved to the fattening unit without being mixed. During the rearing and fattening period pigs were fed a commercial weaner and grower cereal-based diet (Garant Tiernahrung GmbH, Pöchlarn, Austria). During the whole trial, pigs had free access to feed and water.

On day 4 of life, piglets received an intramuscular injection of iron (2.0 mL Ferriphor, Ogris Pharma, Wels, Austria). Male piglets were castrated under general anaesthesia (Azaperon 1 mg/kg body weight (BW); Stresnil, Elanco Animal Health, Bad Homburg, Germany; Ketamin 10 mg/kg BW; Narketan, Vétoquinol Österreich, Vienna, Austria) and received an analgesic treatment (Ketoprofen 3 mg/kg BW; Rifen, Richter Pharma, Wels, Austria) on day 14 of life. Piglets were vaccinated against Mycoplasma hyopneumoniae (Ingelvac MycoFLEX, Boehringer Ingelheim Vetmedica GmbH, Ingelheim/Rhein, Germany) and porcine Circovirus 2 (Ingelvac CircoFLEX, Boehringer Ingelheim Vetmedica GmbH, Ingelheim/Rhein, Germany) on day 21 of life. During the whole experimental trial, the health of the piglets and sows was monitored daily.

### 2.3. Experimental Design and Sample Collection

Twelve hours post partum (pp), a total of 64 clinically healthy, average weight (1.40 ± 0.22 kg body weight (BW)) piglets were selected (8 piglets/litter) and divided into two groups with equal numbers of males and females and balanced for weight. Piglets within the AB group (*n* = 32) received a single intramuscular injection of a vegetable oil-based suspension of ceftiofur crystalline free acid at the recommended dose of 5.0 mg/kg BW (Naxcel, Zoetis Belgium SA, Louvain-la-Neuve, Belgium) on day 0 of life within one hour after the selection. Piglets within the non-treated control group (*n* = 32) received an intramuscular injection of phosphate-buffered saline (0.2 mL/kg BW; NaCl 0.9% B.Braun, B.BRAUN Melsungen AG, Melsungen, Germany). Sows between time of insemination and weaning and pigs throughout the whole experimental period did not receive any treatment with AB. Using swabs, individual fecal samples were collected prior to treatment on day 0 and day 12 of life from all piglets (*n* = 32/group) and on days 28 and 97 of life from 32 pigs (*n* = 16/group). The fecal samples were immediately snap-frozen in liquid nitrogen and stored at −80 °C. The BW of the pigs was measured on days 0 (prior to treatment), 12, 28, 47, and 97 of life prior to faecal sampling. The average daily weight gain for the suckling, postweaning, and early growing period was calculated.

### 2.4. DNA Extraction and Preparation of 16S rRNA Amplicon Libraries

DNA extracts were prepared from 250 mg feces using the PowerSoil DNA isolation kit (MoBio Laboratories, Inc., Carlsbad, CA, USA) with some modifications [19]. After addition of buffer C1, an additional heating step (10 min at 70 °C) was included before bead beating to ensure a proper lysis of bacteria. DNA concentrations were measured on a Qubit 2.0 fluorometer (Life Technologies, Carlsbad, CA, USA) using the Qubit double-stranded DNA (ds DNA) HS assay kit (Life Technologies, Carlsbad, CA, USA). The *16S rRNA* gene was amplified using the primer set 341F (CCTACGGGRSGCAGCAG) and 909R (TTTCAGYCTTGCGRCCGTAC) targeting the V3–V5 hypervariable regions of the *16S rRNA* gene to generate an approximate amplicon size of 510 base pairs (bp). The *16S rRNA* gene PCR, library preparation as well as DNA sequencing of samples was performed by a commercial provider (Microsynth AG, Balgach, Switzerland). Libraries were constructed by ligating sequencing adapters and indices onto purified PCR products using the Nextera XT sample preparation kit (Illumina Inc., San Diego, CA, USA) according to manufacturer’s recommendations. The library preparation included a quality control of the samples and Nextera two-step PCR amplification. For each of the libraries, equimolar amounts were pooled and sequenced on an Illumina MiSeq Personal Sequencer using a 300 bp read length paired-end protocol. The resultant overlapping paired-end reads were stitched and quality-filtered by Microsynth.

### 2.5. Sequence Processing and Analysis

Illumina MiSeq sequencing generated a total of 11,860,700 reads for the 190 samples. The Quantitative Insights Into Microbial Ecology (QIIME) pipeline (version 1.9.1) [20] was used to analyze the *16S rRNA* gene sequences. Fastq files were quality trimmed using the “split_libraries_fastq “script for non-multiplexed Illumina fastq data (phred offset 33). The UCHIME method in the 64-bit version of USEARCH together with the GOLD database (drive5.com) [21,22] were used for filtering of chimeric sequences. A total of 8,098,525 reads that passed the quality control and chimera check. The average number of assembled sequence reads per sample was 42,775 with a median read length of 510 bp. Sequences were clustered into operational taxonomic units (OTU) by UCLUST [21] using the Greengenes reference database (version gg_13_8) [23] and a similarity threshold of 97%. OTUs with less than 10 sequences were removed. In total, 6738 distinct OTUs could be identified for the four sampling time points. Downstream analysis including taxonomy assignment and community metrics analyses (coverage, alpha- and beta-diversity) were constructed in QIIME. Taxonomy was assigned using the SILVA SSU database version 119 [24]. For alpha- and beta-diversity analyses, a rarefaction depth of 5000 sequences per sample was used. Alpha-diversity (Shannon, Simpson) and species richness (Chao 1) analyses were performed using QIIME. For the beta-diversity analysis, the adonis2 function in the vegan R package (V.2.5.5 [25]) in R studio (version 1.2.1335) was used to statistically assess dissimilarity matrices (Bray-Curtis) derived from the fecal microbiome data [25]. The PERMANOVA was applied on the Bray–Curtis distance matrices between factors (litter, sex, AB treatment, day of life, and their multiple interactions) and statistical significance was calculated after 999 random permutations. The dissimilarity matrices were visualized in two-dimensional nonmetric multidimensional scaling (NMDS) ordination plots obtained with the ‘metaMDS’ function in the vegan package [25].

Raw data were submitted to NCBI database (PRJNA574270).

### 2.6. Statistical Analysis

Raw read counts from the tables of OTU abundances were collapsed and compositionally normalized such that each sample sums to 100%. To test for differences in the bacterial taxonomy, taxa appearing in at least 50% of the fecal samples at a relative abundance >0.01% were considered and the relative abundances were analyzed. Bacterial diversity indices, relative abundances at the respective taxonomic ranks, pig’s BW, and average daily gain were tested for normal distribution by the Shapiro–Wilk test with the UNIVARIATE procedure in SAS (Version 9.4, SAS Inst. Inc., Cary, NC, USA). To compare differences between treatment groups (AB versus control), variables (i.e., microbiome data and pig’s performance) were subjected to ANOVA using the MIXED procedure in SAS. Data were analyzed as repeated measures over sampling time points including the fixed effects of sampling day, sex, and treatment as well as their three-way-interaction. Pig within litter was the experimental unit. The degrees of freedom were approximated by the Kenward–Rogers method (ddfm = kr). Results are reported as least-squares means ± standard error of the mean (SEM). Pairwise comparisons among least-square means were computed using the pdiff statement. Differences were considered significant at *p* < 0.05 and trends at 0.05 < *p* ≤ 0.10. Pearson’s correlation analysis between alpha-diversity indices and pig’s BW was additionally performed in SAS. Sparse partial least square regression and relevance network analysis were performed using the mixOmics R package (version 6.8.0; [26]) to integrate data of families (>0.05% of all reads) with BW in order to identify the most influential bacterial taxa on BW development of pigs. Due to the fact that many *Clostridiales* and *Bacteroidales* genera and OTUs were unclassified, we used the next higher level of taxonomy for the establishment of dependencies. Relationships were thereby identified for the individual time points and across the whole experimental period. For this, the ‘network’ function calculated a similarity measure between X and Y variables in a pairwise manner. In the graphs, each X-and Y-variable corresponds to a node, while the edges display the associations between the nodes. The size of the nodes is arbitrary, depending on the name of the variable.

## 3. Results

### 3.1. Body Weight

All pigs remained clinically healthy throughout the trial and no pig died and needed to be removed from the trial due to health or well-being issues. The BW did not differ between the two treatment groups and between male and female pigs on days 0, 12, and 28 of life. On day 97, however, the male pigs of the AB group weighed more compared to the females of the AB group (*p* < 0.05) and male pigs of the control group (*p* < 0.10). In contrast, the opposite was observed in female pigs, where the female AB pigs tended (*p* < 0.10) to have a lower BW compared to the females of the control group on day 97 of life (Figure 1). Average daily weight gain was not different among treatment groups, whereas it was affected by litter (*p* < 0.05) in the suckling and post-weaning period and sex (*p* < 0.05) in the growing period (Appendix A).

### 3.2. Species Richness and Diversity and Community Comparison

For computing alpha-diversity estimators, reads of all samples were normalized to a rarefaction depth of 5000 reads per sample. At this level, rarefaction curves reached an asymptotic shape and diversity was nearly saturated, indicating that the majority of the microbial diversity within the fecal samples has been sufficiently captured. The Chao1 estimator, as measure for species richness, increased over all sampling time points (*p* < 0.05). Likewise, the Shannon diversity index and the Simpson index increased (*p* < 0.05) from day 0 to day 97 of life. On day 97, the females of the control group had a higher species richness (Chao1) compared to the females of the AB group (*p* < 0.05). In addition, females of the control group tended (*p* < 0.10) to have a greater number of species compared to the male control pigs (Figure 2). According to the Shannon and Simpson diversity indices, both control groups harbored a more diverse microbiota than the females of the AB group on day 12 (*p* < 0.05). Furthermore, the females of the control group tended (*p* < 0.10) to have a higher Shannon diversity index than the females of the AB group on day 97. Additionally, strong positive correlations between BW and Shannon index (*r* = 0.71, *p* < 0.001) and Chao1 (*r* = 0.81, *p* < 0.001) were detected.

The PERMANOVA (Bray–Curtis derived distances) showed that the bacterial community structures were similar among sexes, AB treatment and their interaction (Appendix A), whereas differences in bacterial communities were found for litter and day of life. Bray–Curtis derived differences for day of life, sex, treatment, as well as day of life and treatment group were visualized in NMDS plots (Figure 3).

### 3.3. Age-Related Taxonomic Composition of Bacterial Communities

Throughout all sampling time points, 14 different phyla were identified, with different phyla predominating with increasing age of the pigs (Figure 4). On day 0, the predominant phyla were Proteobacteria (86.63% average value) and Firmicutes (13.03% average value) across both sexes. Accordingly, the family *Enterobacteriaceae* (83.25% average value) dominated the microbiota composition on day 0, followed by *Clostridiaceae* (9.90% average value) across sexes. These predominant families were mainly represented by four highly abundant OTUs which accounted for 84% of all reads, including the unclassified *Enterobacteriaceae*-OTU1 (75.92% average value), *Clostridium perfringens*-OTU2 (5.39% average value), unclassified *Enterobacteriaceae*-OTU4 (3.19% average value) and unclassified *Clostridiaceae*-OTU5 (0.43% average value) on day 0 across sexes. 

On day 12, the microbiota composition started to become more diverse, with the relative abundance of the phyla Firmicutes (31.35% average value), Actinobacteria (3.18% average value), and Synergistetes (3.05% average value) being increased (*p* < 0.05) compared to day 0, whereas the relative abundance of Proteobacteria (59.15% average value) was reduced (*p* < 0.05) compared to day 0 across sexes (Figure 4). At family level, the relative abundance of the *Proteobacteria* family *Enterobacteriaceae* (54.14% average value) decreased (*p* < 0.05) but they remained the most abundant family on day 12. Within the phylum Firmicutes, the families *Ruminococcaceae* (12.88% average value), an unclassified *Clostridiales* family (6.03% average value), and *Clostridiaceae* (4.83% average value) were reduced (*p* < 0.05) on day 12 compared to day 0. Moreover, the family *Christensenellaceae* (3.79% average value) was present at a higher (*p* < 0.05) relative abundance on day 12 compared to day 0. Other families appearing on day 12 were the *Synergistaceae* (2.89% average value), *Veillonellaceae* (1.86% average value), *Bifidobacteriaceae* (1.14% average value) and *Coriobacteriaceae* (1.39% average value). At species level, an unclassified *Enterobacteriaceae*-OTU1 (49.25% average value) was the most predominant OTU on day 12.

On day 28, bacteria belonging to the phylum Firmicutes (74.81% average value) were the most abundant, followed by Bacteroidetes (8.11% average value), Proteobacteria (7.87% average value) and Synergistetes (2.50% average value) across sexes (Figure 4). At family level, *Ruminococcaceae* (37.20% average value) was still the most abundant family compared to day 12. An unclassified *Clostridiales* family (12.89% average value), *Christensenellaceae* (11.40% average value), and *Clostridiaceae* (7.63% average value) were other major bacterial families within the phylum Firmicutes on day 28 of life. Furthermore, *BS11* (6.18% average value) and an unclassified *Bacteroidales* family (1.74% average value) were the predominant bacterial families within the phylum of Bacteroidetes, which were more abundant (*p* < 0.05) on day 28 compared to day 12.

On day 97, the Firmicutes (68.79% average value) were less abundant compared to day 28, while the Proteobacteria (22.92% average value) and Tenericutes (1.77% average value) increased (*p* < 0.05) in their abundances compared to day 28 across sexes. Moreover, the Bacteroidetes (3.88% average value) and Synergistetes (0.11% average value) were reduced (*p* < 0.05) on day 97 compared to day 28 (Figure 4). At family level, the *Ruminococcaceae* (24.37% average value) decreased (*p* < 0.05) in their abundance on day 97 compared to day 28, but they still remained the most abundant family. The *Clostridiaceae* (13.07% average value) and an unclassified *Clostridiales* family (13.26% average value) increased (*p* < 0.05) in their abundance on day 97 compared to day 28. Other major families on day 97 were the *Succinivibrionaceae* (15.70% average value), the *Veillonellaceae* (10.94% average value), the *Campylobacteraceae* (2.08% average value), and the *Lactobacillaceae* (1.97% average value). At species level, an unclassified *Ruminococcaceae*-OTU24 (2.41% average value) dominated the family *Ruminococcaceae* on day 97 which was nearly absent on day 0, 12, and 28. By contrast, there was a general decline in those *Ruminococcaceae*-OTUs which were highly abundant on days 12 and 28 (Appendix A).

### 3.4. Effects of the Antibiotic Treatment on Bacterial Microbiota Development

No differences in the GI microbiota were visible among treatment groups and sex on day 0. On day 12, the females of the AB group harbored 1.2 and 1.3-fold more Proteobacteria than the female and male control pigs, respectively (Appendix A). In contrast to that, the female AB pigs showed a 1.7- and 1.9-fold reduction in the relative abundance of Firmicutes compared to the female and male control pigs on day 12. Furthermore, the female control pigs had a 23.6-fold increase in relative abundance of Verrucomicrobia compared to the female AB pigs, whereas no differences existed in males (*p* > 0.05; Appendix A). Within the phylum Proteobacteria, the relative abundance of the family *Enterobacteriaceae* was 1.3- and 1.5-fold higher in females of the AB group compared to female and male control pigs, respectively (Appendix A) on day 12. Additionally, the female control pigs had 4.0- and 3.3-fold more species belonging to the *Campylobacteraceae* than the females and males of the AB group, respectively, on day 12 (Appendix A). Male Moreover, the female AB pigs harbored 2.7-fold less *Desulfovibrionaceae* and 2.2-fold less of an unclassified *Clostridiales* family compared to the female control pigs and again, no differences (*p* > 0.05) were seen between the male pig groups (Appendix A). In addition, female AB pigs comprised 5.1-fold less *Christensenellaceae* compared to the females of the control group on day 12 (Appendix A). Moreover, the AB treatment enriched the most abundant unclassified *Enterobacteriaceae*-OTU1 by 1.5-fold in females of the AB group compared to male control pigs on day 12 (Appendix A). AB pigs showed a 2.6-fold decrease in the relative abundance of the predominant *Campylobacter*-OTU19 compared to male pigs of the control group (Appendix A) on day 12.

On day 28, males of the AB group comprised 1.8-fold more *Bacteroidetes* compared to the male control pigs. Moreover, differences (*p* < 0.05) between the treatment groups were visible within the *Tenericutes*, with the females of the AB group comprising 2.5- and 2.1-fold more *Tenericutes* than the female control pigs and males of the AB group, respectively (Appendix A). Furthermore, males of the AB group had 1.7-fold more bacteria of an unclassified *Clostridiales* family compared to the male control pigs (Appendix A). The female control pigs comprised 2.3-fold less *Christensenellaceae* compared to the male control pigs. Moreover, male pigs of the AB group had 2.3-fold less *Turicibacteraceae* than the male control pigs on day 28 (Appendix A). Furthermore, we could identify differences (*p* < 0.05) for an unclassified *Bacteroidales* family on day 28, which were 3.7-fold less abundant in females of the AB group compared to females of the control group. Similar changes were observed for the family *Helicobacteriaceae*, with female AB pigs harboring 14.9-fold less *Helicobacteriaceae* than the female control pigs on day 28 (Appendix A). Within the family *Ruminococcaceae*, unclassified *Ruminococcaceae*-OTU3, unclassified *Ruminococcaceae*-OTU28 and unclassified *Ruminococcaceae*-OTU46 were the most abundant and differed (*p* < 0.05) among the treatment groups and between males and females (Appendix A) on day 28. Moreover, the dominant unclassified *Christensenellaceae*-OTUs developed differently in response to the AB treatment, for instance, unclassified *Christensenellaceae*-OTU45 and unclassified *Christensenellaceae*-OTU58 being less and more abundant in pigs of the AB group compared to pigs of the control group, respectively, on day 28 (*p* < 0.05; Appendix A). Overall, the detailed treatment effects can be found in Appendix A.

On day 97, females of the AB group showed a 1.8-fold increased relative abundance of the phylum Proteobacteria compared to the female control pigs and the male AB pigs comprised 2.5-fold less Bacteroidetes than male control pigs (Appendix A). In addition, the phylum Tenericutes was 1.5-fold less abundant in AB females compared to female control pigs (Appendix A). Moreover, the female control pigs had a 2.4-fold increased relative abundance of Archaea compared to the females of the AB group (Appendix A). Females of the AB group comprised 1.6-fold more Veillonellaceae compared to the female control pigs (Appendix A) on day 97. In addition, male and female AB pigs had less Lactobacillaceae than the control pigs (Appendix A), whereas only the females of the AB group also had 1.5-fold less Streptococcaceae compared to the female control pigs. The predominating *Proteobacteria*-family *Succinivibrionaceae* was 2.4-fold more abundant in female AB compared to female control pigs, whereas the opposite effect was observed in males of the AB and control groups (Appendix A) on day 97. In addition, the male AB pigs comprised 3.0-fold more *Campylobacteraceae* than the male control pigs (Appendix A), which was mainly due to the 2.3-fold increase in the dominant *Campylobacter*-OTU49 in males of the AB group (Appendix A). At species level, also several dominant *Ruminococcaceae*-OTUs were differently abundant between treatment groups and sexes (Appendix A).

### 3.5. Associations Between Bacterial Families and Host

Due to the relationship between bacterial diversity and BW in pigs, sparse partial least squares regression and relevance networking was used to identify bacterial families that were most discriminative for the BW of pigs across the experimental period (Figure 5A) and separately per day of life. Only the strongest covariation scores for these associations are presented in Figure 5. Accordingly, seven bacterial families were positively associated *(|r|* > 0.6) with the BW of pigs, including *Veillonellaceae*, *Streptococcaceae*, *Lactobacillaceae*, *Succinivibrionaceae,* and unclassified families belonging to RF39 (*Tenericutes*), RF32 (*Proteobacteria*) and GMD14H09 (*Proteobacteria*), across all sampling time points. Only weak relationships (|*r*| = 0.20 to 0.26) between BW and two (day 0 of life) and nine families (day 12 of life). On day 28 of life, *Bifidobacteriaceae*, *Corynebacteriaceae*, an unclassified *Clostridiales* family and *Streptococcaceae* were positively, whereas *Campylobacteraceae* was negatively associated with pig’s BW (*|r|* > 0.4). On day 97 of life, three families including *Lactobacillaceae*, *Elusimicrobioceae* and an unclassified RF39 family were positively related to BW (*r* > 0.35).

## 4. Discussion

Disruptions in the early gut microbiota development may have life-long consequences for the host by affecting the development of the gut and the immune system [27]. Therefore, the present study provides valid data for our increasing understanding of how AB administration early in life, even as a single intramuscular injection, affects the development of the GI microbiota, which may have consequences for the growth performance of pigs later in life. Ceftiofur-mediated alterations in the development of the fecal microbiota composition were evident on sampling days 12, 28, and 97 of life and were different for female and male pigs. Albeit not detected during the suckling phase, the AB administration affected the BW development of the pigs in the long-term (day 97 of life), again differently in females and males. Besides other factors related to host metabolism that probably influenced the BW at d 97 of life, the strong correlations between BW and bacterial species richness (Chao1) and diversity (Shannon and Simpson indices) indicated that the AB-associated decrease in the bacterial alpha-diversity may have been a contributing factor for the depression in BW gain in pigs of the AB and control groups. These data were confirmed by sparse partial least square regression and relevance networking, indicating the reduction or loss of some bacterial taxa (e.g., *Lactobacillaceae* and *Veillonellaceae* across all sampling time points) as a consequence of the ceftiofur injection as potential causative agents for the aberrant BW development of AB-treated pigs. Notably, different bacterial families were identified as the most discriminant for the BW on the individual sampling days, which might have been related to bacterial maturation. 

As reported previously [28,29,30], age and diet were the main factors influencing the development of the fecal bacterial microbiota, with the greatest shift in the phyla Proteobacteria and Firmicutes during the suckling phase. Aside from maturational changes leading to a more diversified fecal microbiota composition from day 0 to 28 of life, introduction of solid feed on day 10 of life likely further promoted an earlier diversification of the fecal microbiota in pigs of the present study compared to other studies in which suckling piglets had no access to creep feed [28,30]. Nevertheless, species richness (Chao1) and diversity (Shannon index) increased over all four sampling time points, supporting that plant-based diets are a major factor for bacterial diversification in the large intestine with increasing age [30]. With respect to specific changes in bacterial taxa, day 0 of life was characterized by early colonizers belonging to *Enterobacteriaceae* and *Clostridiaceae*, whereas *Clostridiaceae*, *Ruminococcaceae*, *Veillonellaceae*, *Christensenellaceae,* and an unclassified *Clostridiales* family dominated the later preweaning fecal microbiomes. As facultative anaerobes, *Enterobacteriaceae* can tolerate the aerobic conditions in the neonatal gut; hence, high fecal abundance of *Enterobacteriaceae* directly after birth was expected in the present study. *Enterobacteriaceae* lower the redox potential in the neonatal gut, supporting the growth of strict anaerobes which appear during the first week of life [31]. Likewise, *Clostridiaceae* are typical early colonizers of the neonatal GI tract and function as precursor for anaerobic bacteria like *Bacteroides* [31]. We did not find a clear predominance of *Bacteroidales* which have been associated with milk-glycan utilization during the suckling period as reported by others [28,29,30]. Instead, *Christensenellaceae* increased in feces until day 28 of life in the present study, indicating their potential role in fermentation of milk sugar, which would be supported by results for stool of lactose-intolerant humans where *Christensenellaceae* increased after the ingestion of galacto-oligosaccharides or dairy-based diets [32]. Moreover, plant carbohydrate-degrading bacteria (e.g., *Ruminocococcaceae*) already played a role during the suckling phase which can be related to the introduction of creep feed from day 10 of life. Many *Veillonella* species utilize primary fermentation metabolites, such as lactate and succinate, as a carbon and energy source for growth [33]. Consequently, their rise during the suckling phase may indicate increased substrate availability via crossfeeding relationships with lactate and succinate producers, such as *ruminococci* and *lactobacilli* [33,34]. Despite the further increase in species richness (Chao1) and diversity (Shannon), differences in the microbiota composition were not as pronounced between days 28 (preweaning) and 97 (postweaning) as it might have been expected from previous work [35], which can be linked to the early introduction of solid feed in the present study. Commonly, weaning implies an abrupt shift from a high-fat, low-carbohydrate milk as the main feed source towards a high-carbohydrate, low-fat cereal-based diet, leading to taxonomic [36] and functional [28] shifts in the gut microbiota which are necessary for adaption to the new feed source. Interestingly, amylolytic *Ruminococcaceae* did not further increase as could be expected for starch- and hemicellulose-degrading bacteria but decreased from day 28 to 97 of life. This may have been in relation to the high easily digestible starch content of the creep feed and the higher intake of creep feed by the piglets in the last days before weaning. Taxa within the family *Succinivibrionaceae* ferment carbohydrates to succinate and acetate [37]. Therefore, their rise together with the rise in plant-derived mono- and disaccharide-consuming and lactate-producing *Lactobacillaceae* may have promoted the increase in *Veillonellaceae* via crossfeeding. Alterations in taxa abundances within *Clostridiaceae* and the unclassified *Clostridiales* family may have modified the primary fermentation metabolite profile, thereby altering the substrate availability for crossfeeding postweaning [33,38]. 

Several rationales may explain the sex-related responses of the microbiome to the ceftiofur including diverging hepatic drug metabolism [39] and subsequent excretion of AB-metabolites via bile in females and males and, although this study was conducted in young prepubertal females, an estrogen-related effect [40]. Accordingly, for a chemically similar AB (i.e., first generation chepalosprine cephradine) a lower absorption and bioavailability has been reported for women compared to men [41]. As these observations were made in sexually mature humans, present results may indicate that sex-related differences in the metabolization of ceftiofur already occur in sexually immature pigs. This might explain the AB-related differences in bacterial taxa abundances between male and female pigs on day 12 of life. For the interpretation of the results on day 28 and 97 of life, it needs to be considered that all male pigs were castrated on day 14, while the females remained intact in our study. The sex-specific treatment-related differences were increasingly pronounced as the piglets grew older, suggesting that the onset of sexual hormone production may play an important role in AB-related shifts of the gastrointestinal microbiota. This is supported by the findings of another study, where sex-associated differences in the GI microbiota in intact and gonadectomized mice were investigated [42]. 

As a parenteral antibiotic, ceftiofur is administered intramuscularly with an effective plasma concentration for at least 158 h according to the product description [43]. Previously, it has been reported that the majority of excreted metabolites in feces is microbiologically inactive [17]. However, the present results strongly emphasize that ceftiofur-metabolites impact the gut microbiota in the short- and long-term. After hepatic biotransformation, ceftiofur-metabolites are subsequently eliminated via urine (68%) and feces (13%) for a time span of 10 days [17,43]. During this time, ceftiofur-metabolites may impact the colonization of the gut microbiota as they reach the GI tract via biliary secretion [17] thereby largely explaining the effects observed on day 12 of life. For instance, the feces of ceftiofur-treated female piglets contained more *Enterobacteriaceae, Coriobacteriaceae,* and *Bifidobacteriaceae* but less *Christensenellaceae* and less of an unclassified *Clostridiales* family on day 12 of life. Since *Enterobacteriaceae* were mostly made up of one single *Escherichia coli*-like OTU and being the most dominant taxa on day 12 of life, especially the increase in this family very probably contributed to the loss of bacterial diversity. Without further phenotyping the present *Escherichia coli* population, it is hard to deduce whether commensal or enteropathogenic strains were promoted. Later in life, the effect on the aforementioned families was less visible, potentially being affected by bacterial changes due to the maturation of the gut microbiota, when the dominance of *Enterobacteriaceae* and *Bifidobacteriaceae* diminished, and due to the lack of sow milk oligosaccharides as bacterial substrate after day 28 of life. However, as other taxa were affected on day 28 and 97 of life, it can be assumed that the early alterations in the gut microbiota due to the ceftiofur administration caused long-lasting shifts in the successional bacterial colonization, having led to the altered bacterial communities on day 28 and 97 of life in our study. More detailed studies on long-term excretion of ceftiofur metabolites may be helpful to explain long-term effects of ceftiofur on the gut microbiome in pigs. Nevertheless, for the interpretation of the data, it should be considered that a pig of 97 days of age is only an adolescent whose bacterial community only begins to stabilize [44]. Against this background, our findings are in line with the effects of parenteral administrated antibiotics (ampicillin and gentamicin) on the fecal microbiota in infants, as seen in a reduction of beneficial bacteria, such as *Bifidobacteriaceae* and *Lactobacillus* [45]. Likewise, we also observed a depression in the *Lactobacillaceae* abundance due to the ceftiofur administration. In contrast to the human study [45], this was a longterm effect in our study, probably as a consequence of an impaired maturation of the microbiota, as this effect was only detectable on day 97 of life. Members of the *Lactobacillaceae* are known to have many health promoting properties [46] including prevention of adhesion of opportunistic pathogens via attaching to the mucus layer in the gut [47]. They also produce lactate which is utilized by other bacteria to convert it into propionate, valerate, and butyrate [48]. Therefore, the reduction in *Lactobacillaceae* may have also altered crossfeeding relationships among bacteria. 

As a broad spectrum AB, ceftiofur targets both, Gram-positive and Gram-negative bacteria [17]. Its bactericidal effect is attributable to its action on the cell wall, basically by disrupting the cell wall synthesis [17]. Especially the differences detected for the microbiota composition on day 12 of life can be directly associated with the AB injection, whereas the effects observed at the later sampling time points may be the consequence of the aberrant early bacterial colonization due to the AB. Growth inhibition of Gram-positive bacteria by ceftiofur included for instance *Christensenellaceae* on day 12 of life, a bacterial family associated with metabolism of several sugars and acetate and butyrate production [49]. However, this taxon has been also associated with lean body mass in humans [50], indicating already a potential link between microbiota development and BW later in life in the present study. Other inhibiting effects on bacterial taxa were more pronounced in females on day 12 of life, clearly pointing at differences in the clearance of the AB via the GI tract. Accordingly, *Lachnospiraceae*, *Campylobacteraceae,* and *Desulfovibrionaceae* were mainly depressed in females but not in males on day 12 of life. In comprising pathobionts [51], a lower abundance of *Campylobacteraceae* and *Desulfovibrionaceae* may be favorable for GI health, especially at this very young age of the piglets. However, a higher mucosal abundance of *Campylobacteraceae* has been linked with an improved feed efficiency in growing pigs [52]. Consequently, a similar relationship may be behind the trend for an increased BW in AB males on day 97 of life, as they also comprised more *Campylobacteraceae* in feces compared to the other three pig groups.

Depressing the GI abundance of *Lachnospiraceae* on day 12 of life may be counterproductive for intestinal health and functioning as butyrate serves as a major energy source for enterocytes and both, propionate and butyrate, modulate mucosal gene expression and exert anti-inflammatory properties [53,54]. Many *Lachnospiraceae* bacteria rely on crossfeeding of primary fermentation metabolites, such as lactate or succinate [48]. Hence, lower *Lachnospiraceae* abundance already indicated alterations in the microbe-to-microbe interactions which may have become more decisive for bacterial abundances between pig groups on day 97 of life. For instance, succinate-producing *Succinivibrionaceae* were largely increased in AB-treated females on day 97 of life which may have resulted in the increase of succinate-metabolizing taxa within *Veillonellaceae* [33]. 

Beside the inhibiting action of ceftiofur on certain Gram-negative bacteria in our study, e.g., *Campylobacter*, the dominant gram-negative family, i.e., *Enterobacteriaceae*, seemed not to be depressed by ceftiofur. In contrast, the AB raised *Enterobacteriaceae* numbers on day 12 of life especially in feces of AB females, indicating that this family may have had a growth advantage potentially due to the bactericidal action of ceftiofur on bacteria with similar substrate preferences. This may have contributed to the observed reduced diversity in female piglets which received the AB on day 12 of life. Albeit *Enterobacteriaceae* comprises many gut commensals, this family is a source of potent opportunistic pathogens in suckling piglets and pigs in the early postweaning period [55]. Stressors like weaning may promote disturbances in the bacterial community and an imbalanced microbiota enables opportunistic pathogens to become pathogenic [56,57], potentially rendering those piglets more susceptible to enteric disease after weaning. The lack of an effect of ceftiofur on *Enterobacteriaceae* might be explained by the fact that *Enterobacteriaceae* may have developed resistance to cephalosporins [14].

## 5. Conclusions

In conclusion, the present results demonstrate that an early single parenteral injection of ceftiofur 12 h pp induced changes in the bacterial colonization in the short- and long-term. Moreover, results indicated sex-related differences in the AB response, suggesting differences in the metabolism of the AB and intestinal secretion of AB metabolites as a potential underlying mechanism. Especially the loss of bacterial diversity in female pigs receiving the AB may have been a contributing factor for the decreased BW of these females on day 97 of life. Overall, the present findings emphasize the importance of proper and restrictive use of antibiotics in neonatal pigs in order to prevent long-term disturbances of the gut ecosystem and lifelong performance.

## Figures and Tables

**Figure 1 animals-10-00017-f001:**
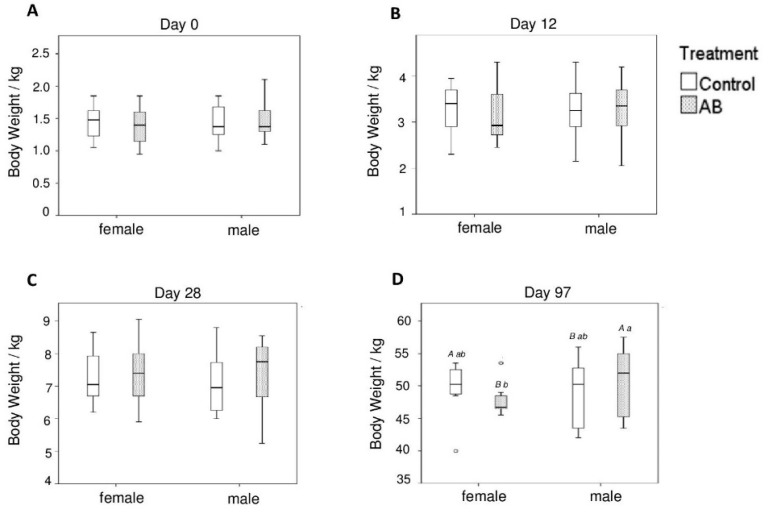
Body weight of female and male pigs either receiving the antibiotic (AB) or control treatment on (**A**) day 0, (**B**) day 12, (**C**) day 28, and (**D**) day 97 of life. Different capital letters indicate a tendency (*p* < 0.1) and different lowercase letters indicate significant differences (*p* < 0.05).

**Figure 2 animals-10-00017-f002:**
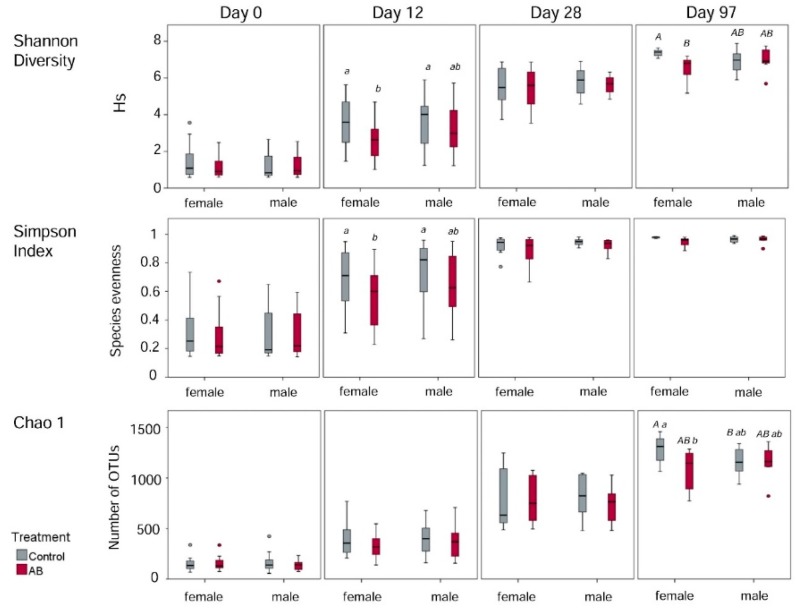
Alpha diversity estimations in female and male pigs either receiving the antibiotic (AB) or control treatment on day 0, 12, 28, and 97 of life. Different capital letters indicate a tendency (*p* < 0.1) and different lowercase letters indicate significant differences (*p* < 0.05).

**Figure 3 animals-10-00017-f003:**
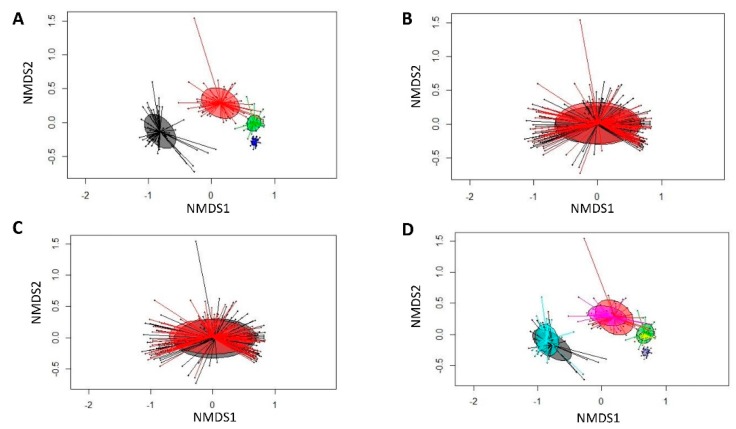
Nonmetric multidimensional scaling (NMDS) plot of pairwise Bray–Curtis dissimilarities between bacterial communities (rel. abundance >0.01%). (**A**) Sampling time point (black, day 0; red, day 12; green, day 28; dark blue, day 97); (**B**) sex (black, female pigs; red, male pigs); (**C**) treatment (control―black circle, antibiotic treatment (AB)―red circle); and (**D**) sampling time point and treatment group (black, day 0: control; turquoise, day 0: AB; red, day 12: control; pink, day 12: AB; green, day 28: control; yellow, day 28: AB; dark blue, day 97: control; light blue, day 97: AB).

**Figure 4 animals-10-00017-f004:**
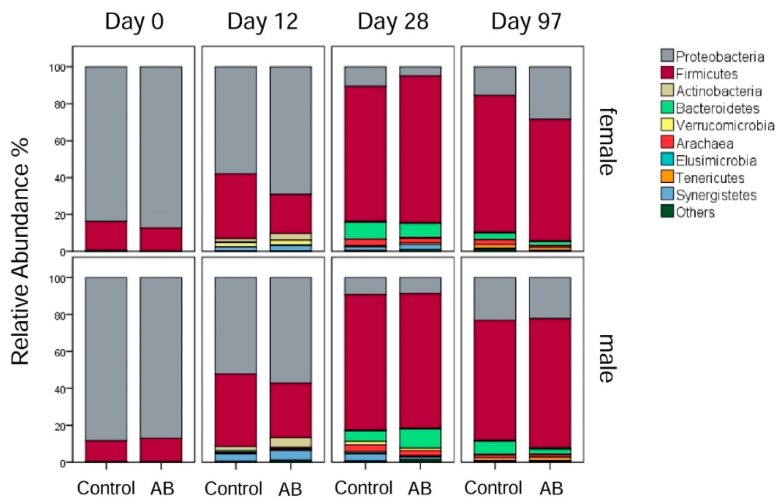
Relative abundances (% of total reads) of bacterial phyla in female and male pigs receiving either the antibiotic (AB) or control treatment on day 0, 12, 28, and 97 of life.

**Figure 5 animals-10-00017-f005:**
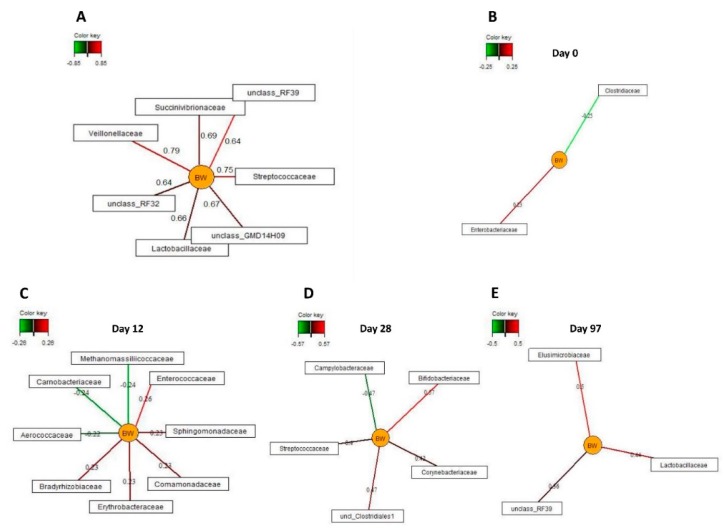
Associations between the relative abundance of the most discriminative bacterial families (>0.05% of total reads) and body weight in female and male pigs receiving either the antibiotic (AB) or control treatment on day 0, 12, 28 and 97 of life. The associations were established separately using sparse partial least squares regression and relevance networking. The networks are displayed graphically as nodes (parameters) and edges (biological relationship between nodes). The edge color intensity indicates the level of the association: red = positive, and green = negative. Only the strongest pairwise associations were projected. (**A**) Relationships across all sampling time points (|*r*| > 0.6); (**B**) relationships on day 0 of life (|r| > 0.2); (**C**) relationships on day 12 of life (|*r*| > 0.2); (**D**) relationships on day 28 of life (|*r*| > 0.4); and (**E**) relationships on day 97 of life (*r* > 0.35).

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
