# Peer review of "Early Parenteral Administration of Ceftiofur has Gender-Specific Short- and Long-Term Effects on the Fecal Microbiota and Growth in Pigs from the Suckling to Growing Phase"

_animals, 2019, doi:10.3390/ani10010017_

Round 1

Reviewer 1 Report

The manuscript significantly improves and the authors positively implement it according to the review comments. I only have some minor comments:

- since permanova and Adonis test were performed on the bray –Curtis distance matrix I would suggest visualizing the same matrix in FIGURE 3 instead of the UNifrac distance matrix. Or does this choice have a particular reason?

- The authors implemented the manuscript including the sPLS analysis between the microbial data at genus level and the piglets’ BW. But it not clears to me if the PLS was a carried out at each time-points in which the data were available or if the PLS was a multilevel PLS. How did the authors manage that the data on piglets were repeated per time? Because the optimal and beneficial bacteria may different per piglets age (and their maturation) I would suggest analyzing the relationship between microbial and BW data separately at each time point (or at least at the most important time points).

Author Response

Response to Reviewer 1

Reviewer #1 (Comments for the Author):

The manuscript significantly improves and the authors positively implement it according to the review comments. I only have some minor comments:

Authors: Thank you for the further helpful comments.

Specific Comments:

Since permanova and Adonis test were performed on the bray –Curtis distance matrix I would suggest visualizing the same matrix in FIGURE 3 instead of the UNifrac distance matrix. Or does this choice have a particular reason?

Authors: Thank you for asking. We provided the PERMANOVA based on pairwise Bray-Curtis dissimilarities when revising our manuscript; however, we kept the illustration using Unifrac distances. We changed FIGURE 3 in such a way that now the pairwise Bray-Curtis dissimilarities are illustrated in a nonmetric multidimensional scaling (NMDS) plot.

The authors implemented the manuscript including the sPLS analysis between the microbial data at genus level and the piglets’ BW. But it not clears to me if the PLS was a carried out at each time-points in which the data were available or if the PLS was a multilevel PLS. How did the authors manage that the data on piglets were repeated per time? Because the optimal and beneficial bacteria may different per piglets age (and their maturation) I would suggest analyzing the relationship between microbial and BW data separately at each time point (or at least at the most important time points).

Authors: Thank you for this comment. Of course, the relationships between the microbial community and BW could be assessed separately for each time point. We modified this analysis accordingly and provided the respective relevance networks in Figure 5.

Reviewer 2 Report

This paper described the “Early parenteral administration of ceftiofur has gender-specific short- and long term effects on the faecal microbiota and growth in pigs from the suckling to growing phase”. The authors have improved the manuscript based on my comments. However, I could understand that ceftiofur has a negative effect on Shannon diversity and Simpson index on day 12 of female piglets. But the Shannon diversity and Simpson index of female piglets were recovered on day 28, implying that some mechanisms were produced in response to the reduced Shannon diversity and Simpson index caused by ceftiofur treatment. On day 97, the Shannon diversity was reduced in ceftiofur-treated female piglets again. Whether some dominant species (good or bad microbes, such as Lactobacillaceae) is altered in response to ceftiofur treatment at the early stage and do not recover at the late stage in female piglets. It would be better to show these results to understand the long term effects of ceftiofur on the faecal microbiota in pigs.

Author Response

Reviewer #2 (Comments for the Author):

This paper described the “Early parenteral administration of ceftiofur has gender-specific short- and long term effects on the faecal microbiota and growth in pigs from the suckling to growing phase”. The authors have improved the manuscript based on my comments.

Authors: Thank you.

Specific Comments:

However, I could understand that ceftiofur has a negative effect on Shannon diversity and Simpson index on day 12 of female piglets. But the Shannon diversity and Simpson index of female piglets were recovered on day 28, implying that some mechanisms were produced in response to the reduced Shannon diversity and Simpson index caused by ceftiofur treatment. On day 97, the Shannon diversity was reduced in ceftiofur-treated female piglets again. Whether some dominant species (good or bad microbes, such as Lactobacillaceae) is altered in response to ceftiofur treatment at the early stage and do not recover at the late stage in female piglets. It would be better to show these results to understand the long term effects of ceftiofur on the faecal microbiota in pigs.

Authors: Thank you for your comment. Indeed, we observed ceftiofur- related changes within different taxa at different time points during the trial. As you noted, there seems to have been a certain “recovery” of the faecal microbiota. However, this may be rather seen in relation to the maturation of the microbiota. The question is whether part of this “recovery” was linked to the introduction of creep feed. However, because all piglet groups received the same feeding regime, this needs to be investigated more closely in a follow-up study. The detailed changes in the most abundant families can be found in Supplemental Table 4. We modified the discussion section to better demonstrate that changes in the faecal microbiota were different on the consecutive sampling days (L451-461).

Round 2

Reviewer 1 Report

The authors fully revised the manuscript according to my suggestions. I would consider it appropriate for the publication.

This manuscript is a resubmission of an earlier submission. The following is a list of the peer review reports and author responses from that submission.

Round 1

Reviewer 1 Report

The manuscript of Ruczizka et al investigates the long and short term effect of parenteral administration of ceftiofur on the faecal microbiota and growth in female and male piglets. The topic is quite relevant since new strategies to improve piglets’ robustness by favouring the intestinal eubiosis can help to reduce the use of antibiotics in animal production. Nevertheless, the study poses some concerns. Below there some specific comments, which need clarification before the acceptance of the manuscript:

Title:

Since the authors found that the antibiotic administration can vary according to gender, it may be mentioned in the title.

Materials and methods section:

The results report the differenced in piglets’ body weight, however, in the material and methods section, the collection of those data and their analysis are not described. Please include it. 

Statistical section: the statistical model used beta diversity index should be described in this section, while only the univariate approach used for the OTUs abundancy and alpha indices have been included by the authors. Furthermore, it is not clear if the litter of origin was considered in the statistical model. The sows’ microbiota, as well as her health status and colostrum and milk composition, may have affected the microbial development of their offspring.

Since the age of animals is known to influence the development and composition of piglets fecal microbiota, the authors could consider investigating the effect of treatment, sex and litter at each age in order to evidence the effect of the others factors. The permutational multivariate analysis of variance (PERMANOVA) can be performed using the “adonis” function in order to assess the community differences between groups.

Results section:

L 270: “Effects of the antibiotic treatment on bacterial microbiota development”. Since the results in this section have been presented by the authors per gender I suggest to include this aspect in the title. Furthermore, I suggest the authors organize the description of the results by age and taxonomy level. This organization is already partially present, and taxonomy level should be highlighted.

Discussion section:

L 340-343: this appears more speculation. The correlation between body weight and the reduction of Lactobacillaceae and increase of Campylobacteriaceae due to the ceftiofur should be statistically tested. A potential model to disclose this can be the multivariate approach PLS.

L414-416 This sentence it speculation and no correlation between Lactobacillaceae and other bacterial taxa have been provided to support this hypothesis.

Author Response

Response to Reviewer 1

Reviewer #1 (Comments for the Author):

The manuscript of Ruczizka et al investigates the long and short term effect of parenteral administration of ceftiofur on the faecal microbiota and growth in female and male piglets. The topic is quite relevant since new strategies to improve piglets’ robustness by favouring the intestinal eubiosis can help to reduce the use of antibiotics in animal production. Nevertheless, the study poses some concerns. Below there some specific comments, which need clarification before the acceptance of the manuscript.

Author: Many thanks for your review. Below you will find our answers and additional information. Additional changes in the revised manuscript are based on the comments of the other reviewer.

Specific Comments:

Title:

Since the authors found that the antibiotic administration can vary according to gender, it may be mentioned in the title.

Authors: Thanks for this comment. We modified the title accordingly to highlight gender-specific differences (L 3)

Materials and methods section:

The results report the differenced in piglets’ body weight, however, in the material and methods section, the collection of those data and their analysis are not described. Please include it.

Authors: There must have been a misunderstanding. The BW measurement and analysis of the data were mentioned in the original manuscript. However, we modified these parts to provide more details about the measurement and data analysis. (L 129-131, L 178-180).

Statistical section: the statistical model used beta diversity index should be described in this section, while only the univariate approach used for the OTUs abundancy and alpha indices have been included by the authors. Furthermore, it is not clear if the litter of origin was considered in the statistical model. The sows’ microbiota, as well as her health status and colostrum and milk composition, may have affected the microbial development of their offspring. Since the age of animals is known to influence the development and composition of piglets fecal microbiota, the authors could consider investigating the effect of treatment, sex and litter at each age in order to evidence the effect of the others factors. The permutational multivariate analysis of variance (PERMANOVA) can be performed using the “adonis” function in order to assess the community differences between groups.

Authors: Thank you for rising this important point. We considered the effect of treatment, sex and litter at each age in the ANOVA. We added results for the PERMANOVA, providing the detailed results for these effects with respect to the beta-diversity (L 164-172 and L 238-240).

Results section:

L 270: “Effects of the antibiotic treatment on bacterial microbiota development”. Since the results in this section have been presented by the authors per gender I suggest to include this aspect in the title. Furthermore, I suggest the authors organize the description of the results by age and taxonomy level. This organization is already partially present, and taxonomy level should be highlighted.

Authors: Thank you for this suggestion. We tried to modify our Results section accordingly.

Discussion section:

L 340-343: this appears more speculation. The correlation between body weight and the reduction of Lactobacillaceae and increase of Campylobacteriaceae due to the ceftiofur should be statistically tested. A potential model to disclose this can be the multivariate approach PLS.

Authors: Thank you for pointing this out. We used sparse partial least square regression to identify relationships between the most influential bacterial families and pig’s BW (L 355-362 and L 385-388)

L414-416 This sentence it speculation and no correlation between Lactobacillaceae and other bacterial taxa have been provided to support this hypothesis

Authors: Thank you for this comment; this sentence has been modified in order to not over-interpret our results (L 467-468)

Reviewer 2 Report

This study evaluated the effect of ceftiofur treatment period on the fecal microbiota of piglets. And the result of body weight on gender differences is interesting.
The manuscript is well written and I think there is no problem, but the resolution of the figures is too low. Increasing the resolution of the figures will not inconvenience readers.

Author Response

Response to Reviewer 2

Reviewer #2 (Comments for the Author):

This study evaluated the effect of ceftiofur treatment period on the fecal microbiota of piglets. And the result of body weight on gender differences is interesting.

The manuscript is well written and I think there is no problem, but the resolution of the figures is too low. Increasing the resolution of the figures will not inconvenience readers.

Authors: Many thanks for your review of our paper. We are grateful for your comment and hopefully could sufficiently improve the quality of our figures.

Additional changes in the revised manuscript are based on the comments of the other reviewers.

Reviewer 3 Report

This paper described the “Early parenteral administration of ceftiofur has short and longterm effects on the faecal microbiota and growth in pigs from the suckling to growing phase”. Some issues need to be further verified. The comments are below:

Previous study has shown that the fecal microbiota profiles were different in response to different antibiotics (Zeineldin et al., 2018). In the present study, piglets were received a single intramuscular injection of ceftiofur crystalline free acid on day 0 and individual fecal samples were collected prior to treatment on day 0, 12, 28 and 97. It has been mentioned that the ceftiofur is administered intramuscularly with an effective plasma concentration for at least 158 hours. How is the change of ceftiofur metabolites in urine and feces during the different time periods? This result could provide the information to support the short and longterm effects of ceftiofur on pigs. Only bacterial evenness (Shannon and Simpson) was affected by ceftiofur treatment on day 12 of female piglets and then recovered on day 28. It is difficult to understand the effect of ceftiofur on fecal microbiota from day 12 to 28. And then bacterial richness (Chao 1) was reduced in ceftiofur-treated female piglets on day 97. It should be provided other evidence to show that ceftiofur has a short and longterm effects on fecal microbiota of piglets, for example fecal or blood metabolome. The correlation analysis should be performed based on bacteria species, at least genus level since the results of richness and evenness are not informative. In addition, the results of mortality, daily gain, daily intake, and feed conversion ratio should be presented.

Author Response

Response to Reviewer 3

Reviewer #3 (Comments for the Author):

This paper described the “Early parenteral administration of ceftiofur has short and longterm effects on the faecal microbiota and growth in pigs from the suckling to growing phase”. Some issues need to be further verified. The comments are below:

Authors: Many thanks your helpful comments. Below you will find our answers and additional information. Additional changes in the revised manuscript are based on the comments of the other reviewers.

Specific Comments:

Previous study has shown that the fecal microbiota profiles were different in response to different antibiotics (Zeineldin et al., 2018). In the present study, piglets were received a single intramuscular injection of ceftiofur crystalline free acid on day 0 and individual fecal samples were collected prior to treatment on day 0, 12, 28 and 97. It has been mentioned that the ceftiofur is administered intramuscularly with an effective plasma concentration for at least 158 hours. How is the change of ceftiofur metabolites in urine and feces during the different time periods? This result could provide the information to support the short and longterm effects of ceftiofur on pigs.

Authors: Thank you very much for asking this important question. According to Beconi-Barker et al. (1996) and Hornish and Kotarski (2002), the majority of metabolites which are excreted via urine (68%) and faeces (13%), are metabolically inactive. Furthermore, ceftiofur is unstable in the environment because of extensive hydrolytic and photolytic mechanisms. Therefore we can exclude an resumption of active metabolites by the piglets.

To our knowledge, there is no publication regarding how long or which pathways underlie the ceftiofur-metabolites excretion after a single injection of a long-acting formulation of ceftiofur because Hornish and Kotarski (2002) and Beconi-Barker et al. (1996) focused on single and daily injections.

According to the product description (https://ec.europa.eu/health/documents/community-register/2011/20110614104728/anx_104728_en.pdf), approximately 60% (urine) and 15% (faeces) of the administered dose is excreted within 10 days. We modified the respective part of the Discussion (L 450-458).

Only bacterial evenness (Shannon and Simpson) was affected by ceftiofur treatment on day 12 of female piglets and then recovered on day 28. It is difficult to understand the effect of ceftiofur on fecal microbiota from day 12 to 28.

Authors: Thank you for your comment. As mentioned in our reply to the previous comment, our results demonstrate that even a small amount of metabolites is sufficient for impairing the maturation of the gut microbiota directly after birth with subsequent chances later in life. Based on the product description and on the effective plasma concentration of the used product, metabolites should be only excreted for a time period of 10 days. However, the present data show the necessity to investigate ceftiofur metabolism for a longer time period. (L 450-458)

And then bacterial richness (Chao 1) was reduced in ceftiofur-treated female piglets on day 97.

Authors: Thank you for mentioning this. This was a long-term effect in our study, assumed as a consequence of an impaired maturation of the microbiota.

It should be provided other evidence to show that ceftiofur has a short and longterm effects on fecal microbiota of piglets, for example fecal or blood metabolome.

Authors: Thank you for this suggestion. We will consider this for our next experiments in relation to ceftiofur administration.

The correlation analysis should be performed based on bacteria species, at least genus level since the results of richness and evenness are not informative.

Authors: Thank you. We added sparse partial least square regression and relevance networking to provide more detailed information on potential relationships between BW development and the most influential bacteria (L 189-196; L 355-362).

In addition, the results of mortality, daily gain, daily intake, and feed conversion ratio should be presented.

Authors: Thanks for your comment. The pigs’ health was monitored daily. During the whole trial no piglet died. We added this information in L 115-116 and L 199-200. Since the pigs had free access to feed and were group-housed, individual feed intake measurements were unfortunately not possible. We added the average daily weight gain data in the results section (L 205-207).
